# Adaptive sampling-based optimization of quantics tensor trains for noisy functions: Applications to quantum simulations

**Kohtaroh Sakaue[1]★, Hiroshi Shinaoka[1] and Rihito Sakurai[1,2]†**

**1** Department of Physics, Saitama University, Saitama 338-8570, Japan
**2** Department of Physics, The University of Tokyo, Tokyo 113-0033, Japan

★ spikekohtaroh@gmail.com , † sakurairihito@gmail.com

## Abstract

Tensor cross interpolation (TCI) is a powerful technique for learning a tensor train (TT) by adaptively sampling a target tensor based on an interpolation formula. However, when the tensor evaluations contain random noise, optimizing the TT is more advantageous than interpolating the noise. Here, we propose a new method that starts with an initial guess of TT and optimizes it using non-linear least-squares by fitting it to measured points obtained from TCI. We use quantics TCI (QTCI) in this method and demonstrate its effectiveness on sine and two-time correlation functions, with each evaluated with random noise. The resulting QTT exhibits increased robustness against noise compared to the QTCI method. Furthermore, we employ this optimized QTT of the correlation function in quantum simulation based on pseudo-imaginary-time evolution, resulting in ground-state energy with higher accuracy than the QTCI or Monte Carlo methods.

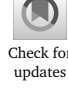

# 1  Introduction

Tensor networks have emerged as a powerful technique for efficiently representing state vectors with exponentially large dimensions, originating from quantum many-body physics. In recent years, their applications have expanded to diverse fields, including image compression [1], machine learning [2,3], solving partial differential equations [4–9], chemical master equation [10], quantum field theory [11–13], and finance [14,15]. The success of tensor networks in these domains can be attributed to the inherent low-rank structure of the target tensors.

This hidden structure enables the efficient compression of high-dimensional tensor objects with numerous elements and computations using a key technique called tensor cross interpolation (TCI) [12,16–18]. TCI has proven to be an effective compression method that builds tensor trains (TTs) by adaptively sampling target tensors with a low-rank structure according to specific rules. Unlike singular value decomposition, which requires access to full tensors, TCI only needs a subset of the tensors, making it highly advantageous. TCI offers a promising approach for solving high-dimensional problems with low-rank structures and can potentially serve as an alternative to the Monte Carlo method [12].

However, when TCI is applied to tensors whose entries are affected by random noise during evaluation, its effectiveness becomes uncertain. This is because TCI interpolates the noise at the chosen points, leading to overfitting. This situation is particularly relevant in practical applications, such as quantum computing or Monte Carlo simulations, where statistical errors are unavoidable when measuring expectation values. In quantum computing, which is the main application of this study, shot noise arises from the finite number of measurements needed to

compute the expectation values of an observable. This noise can lead to inaccuracies in the tensor elements and subsequently affect the quality of the TT approximation obtained through TCI. Therefore, the presence of noise in tensor evaluations raises questions about the robustness and reliability of TCI in practical scenarios. A recent study [19] has investigated the effect of noise in TCI, while previous research has employed an error-mitigated technique such as supervised machine learning, using qubit measurement results as training data in quantum state tomography with TCI [20].

Here, we propose a new method for optimizing TTs that is more advantageous than interpolating noise on chosen tensors. Specifically, we start with an initial guess of TT and optimize it by fitting it to the measured points obtained from TCI by using the non-linear least-squares method. These measured points served for fitting collectively capture the global structure of the target function. This optimization process allows us to mitigate the effect of noise and get the optimized TT that correctly approximates noise-free functions. To demonstrate the effectiveness of our proposed method, we combine it with quantics TCI (QTCI) [21] and apply it to two different functions with continuous variables: a sine function and a two-time correlation function. In both cases, we show that our method achieves higher precision compared to the QTCI method (see Figs. 3 and 7). Finally, we apply our proposed method to quantum simulation based on pseudo-imaginary-time evolution for calculating the ground-state energy. Our proposed method yields a more accurate ground-state energy than those obtained using QTCI or Monte Carlo methods (see Fig. 6). Our results demonstrate that our proposed method enables robust learning TTs from functions under the influence of noise.

This paper is organized as follows: Section 2 introduces the TT and quantics TT. Section 3 introduces TCI for building TTs by adaptive sampling of target tensors. Section 4 introduces our proposed method for optimizing TTs for functions affected by random noise in function evaluations. Section 5 presents a quantum simulation based on a pseudo-imaginary-time evolution algorithm to compute ground-state energy, integrating our proposed method and comparing the results with conventional approaches. Section 6 summarizes this study and discusses future directions.

## 2 Quantics tensor train

We introduce the TT and quantics tensor train (QTT) [22–24], suitable for compressing functions with continuous variables.

### 2.1 Tensor train

A $\mathcal{L}$ th-order tensor, $F_{\sigma_1\sigma_2...\sigma_{\mathcal{L}}}$, where each local index $\sigma_l$ (for $l = 1, ..., \mathcal{L}$) has local dimension $d$, can be decomposed into a TT. The TT representation is a type of tensor network with a one-dimensional structure. A TT decomposition is achieved by applying matrix decomposition techniques [16, 25]:

$$
\begin{aligned}
F_{\sigma_1\sigma_2...\sigma_{\mathcal{L}}} &\simeq \tilde{F}_{\sigma_1\sigma_2...\sigma_{\mathcal{L}}} \\
&= \sum_{\alpha_1=1}^{\chi_1} \cdots \sum_{\alpha_{\mathcal{L}-1}=1}^{\chi_{\mathcal{L}-1}} [F_1]_{1\alpha_1}^{\sigma_1} \cdots [F_l]_{\alpha_{l-1}\alpha_l}^{\sigma_l} \cdots [F_{\mathcal{L}}]_{\alpha_{\mathcal{L}-1}1}^{\sigma_{\mathcal{L}}} \\
&\equiv [F_1]_{\sigma_1} \cdots [F_l]_{\sigma_l} \cdots [F_{\mathcal{L}}]_{\sigma_{\mathcal{L}}} ,
\end{aligned}
\tag{1}
$$

where the third-order tensor $[F_l]_{\alpha_{l-1}\alpha_l}^{\sigma_l}$ has dimensions $\sigma_l \times \chi_{l-1} \times \chi_l$, where $\alpha_l$ is the virtual bond index with dimension $\chi_l$, referred to as the bond dimension. The maximum bond dimension is denoted by $\chi$. If the bond dimension $\chi_l$ remains constant with increasing the number

of tensors $\mathcal{L}$, the tensor $F_{\sigma_1\sigma_2...\sigma_\mathcal{L}}$ has a low-rank structure. The tilde in $\tilde{F}_{\sigma_1\sigma_2...\sigma_\mathcal{L}}$ indicates TT compression, and this notation is used hereafter.

## 2.2 Quantics tensor train

To introduce QTT, we consider a univariate function $f(x)$ defined on $x \in [0,1)$. Here, we discretize the continuous variable $x$ by setting up $d = 2^\mathcal{R}$ equally spaced grids on the $x$-axis. This discretization allows us to obtain a tensor representation of $f(x)$ as a vector $F_\sigma$ with $d$ elements indexed by $\sigma$:

$$F_\sigma = f(x(\sigma)). \tag{2}$$

In QTT, the discrete variable $x$ is encoded in binary code:

$$x(\sigma_1,\ldots,\sigma_\mathcal{R}) = \sum_{r=1}^{\mathcal{R}} \frac{\sigma_r}{2^r}, \qquad \sigma_r \in \{0,1\}, \tag{3}$$

where $\sigma_r$ represents bits indicating different length scales $2^{-r}$, and $\mathcal{R}$ denotes the number of bits. Next, we reshape this vector $F_\sigma$ into an $R$-order tensor of $2 \times \cdots \times 2$:

$$f(x) = f(x(\sigma_1\sigma_2,\ldots,\sigma_\mathcal{R})) = F_{\sigma_1\sigma_2\cdots\sigma_\mathcal{R}} = F_{\boldsymbol{\sigma}}. \tag{4}$$

We assume that the correlation between significantly different length scales is weak. To take advantage of this weak correlation structure, we employ the TT decomposition, as defined in Eq. (1), to compress the tensor as follows:

$$F_{\boldsymbol{\sigma}} = F_{\sigma_1\sigma_2...\sigma_\mathcal{R}} \approx \tilde{F}_{\boldsymbol{\sigma}} = [F_1]_{\sigma_1}[F_2]_{\sigma_2}\cdots[F_\mathcal{R}]_{\sigma_\mathcal{R}}. \tag{5}$$

QTT can also be extended to a multivariate function $f(\boldsymbol{x})$ with real continuous variables $\boldsymbol{x} = (x_1,\ldots,x_n)$ defined in $[0,1)$. Each variable $x_l(l=1,\ldots,n)$ is discretized and encoded in binary code:

$$x_l(\sigma_{l1},\ldots,\sigma_{l\mathcal{R}}) = \sum_{r=1}^{\mathcal{R}} \frac{\sigma_{lr}}{2^r}, \qquad \sigma_{lr} \in \{0,1\}, \qquad l = 1,\ldots,n. \tag{6}$$

In multivariate functions, we perform scale separation, which involves arranging indices representing the same length scales to be adjacent. This yields an $n\mathcal{R}$-order tensor $F_\sigma$ of $f(\boldsymbol{x})$. We decompose $F_\sigma$ as follows using TT decomposition:

$$\begin{aligned} F_{\boldsymbol{\sigma}} &= F_{\sigma_{11},\ldots,\sigma_{n1},\ldots,\sigma_{1\mathcal{R}},\ldots,\sigma_{n\mathcal{R}}} \\ &= f(x_1(\sigma_{11},\ldots,\sigma_{1\mathcal{R}}), x_2(\sigma_{21},\ldots,\sigma_{2\mathcal{R}}),\ldots,x_n(\sigma_{n1},\ldots,\sigma_{n\mathcal{R}})) \\ &\simeq \tilde{F}_{\boldsymbol{\sigma}} = [F_1]_{\sigma_{11}}\cdots[F_{n\mathcal{R}}]_{\sigma_{n\mathcal{R}}}. \end{aligned} \tag{7}$$

As a concrete case of a bivariate function $f(x,y)$ defined for normalized variables $x,y \in [0,1)$, we decompose $F_\sigma$ into $\tilde{F}_\sigma$ as follows:

$$\begin{aligned} F_{\boldsymbol{\sigma}} &= F_{\sigma_{11},\sigma_{21},\ldots,\sigma_{1\mathcal{R}}\sigma_{2\mathcal{R}}} \\ &= f(x(\sigma_{11},\ldots,\sigma_{1\mathcal{R}}), y(\sigma_{21},\ldots,\sigma_{2\mathcal{R}})) \\ &\simeq \tilde{F}_{\boldsymbol{\sigma}} = [F_1]_{\sigma_{11}}[F_2]_{\sigma_{21}}[F_3]_{\sigma_{12}}[F_4]_{\sigma_{22}}\cdots[F_{2\mathcal{R}-1}]_{\sigma_{1\mathcal{R}}}[F_{2\mathcal{R}}]_{\sigma_{2\mathcal{R}}}. \end{aligned} \tag{8}$$

### 2.2.1 Examples of QTT

For some analytic functions, their exact bond dimensions in QTT are known. The simplest example is an exponential function. For the exponential function $f(x) = e^{\lambda x}$, $F_{\sigma}$ can be obtained in QTT as shown in Eq. (9):

$$F_{\sigma} = \prod_{l=1}^{\mathcal{R}} e^{\lambda \sigma_l / 2^l} , \tag{9}$$

where $F_{\sigma}$ can be written as the direct product of $M_l = e^{\lambda \sigma_l / 2^l}$ $(l = 1, \ldots, \mathcal{R})$, which indicates that the QTT has a exact bond dimension of 1.

For another example, the sine function $f(x) = \sin(\lambda x)$, being exactly equal to the subtraction of two exponential functions $e^{i\lambda x}$ and $e^{-i\lambda x}$, is known to have a bond dimension of 2 in QTT [23, 24].

## 3 Tensor train learning algorithm

We introduce a tensor train learning algorithm to build TTs from target tensors: tensor cross interpolation.

### 3.1 Tensor cross interpolation

Tensor cross interpolation (TCI) is a method for building a TT by adaptively sampling a subset of a tensor $F_{\sigma}$ with a low-rank structure [12, 16–18]. TCI is particularly effective when the number of elements of the target tensor is substantially large because it only requires some tensor elements to build the TT. Thus, TCI adaptively selects sampling points to learn the tensor network, which can be regarded as active machine learning. In particular, combining QTT and TCI is called quantics tensor cross interpolation (QTCI) [21], allowing for the efficient compression of functions with continuous variables and computations. TCI is a quasi-optimal algorithm in the maximum norm, as given by

$$\epsilon_{\text{TCI}} = \frac{|F_{\sigma} - \tilde{F}_{\text{TT}}|_{\max}}{|F_{\sigma}|_{\max}} , \tag{10}$$

where $F_{\sigma}$ is the original function, and $\tilde{F}_{\text{TT}}$ is the estimated TT obtained by TCI [18]. The denominator is a normalization factor, which is the estimated absolute value of the function $F_{\sigma}$, denoted as $|F_{\sigma}|_{\max}$, and is used to handle cases where $F_{\sigma}$ might include zero values for numerical stability. In this paper, we perform TCI by setting the maximum bond dimension of $\tilde{F}_{\text{TT}}$, instead of specifying a tolerance as in Eq. (10).

TCI is an interpolation formula, meaning that at the selected interpolation point $\sigma$, $F_{\sigma}$ strictly matches $\tilde{F}_{\text{TT}}$ as

$$\tilde{F}_{\text{TT}} = F_{\sigma} . \tag{11}$$

Here, we define measured points as the set of all indices and corresponding tensor values sampled during the execution of TCI. Interpolation points are a subset of these measured points used to build the TT through TCI.

### 3.2 Singular value decomposition

In this study, we use singular value decomposition (SVD) to reduce further the bond dimension of TTs obtained from TCI. SVD can decompose a given TT into another TT with an optimal

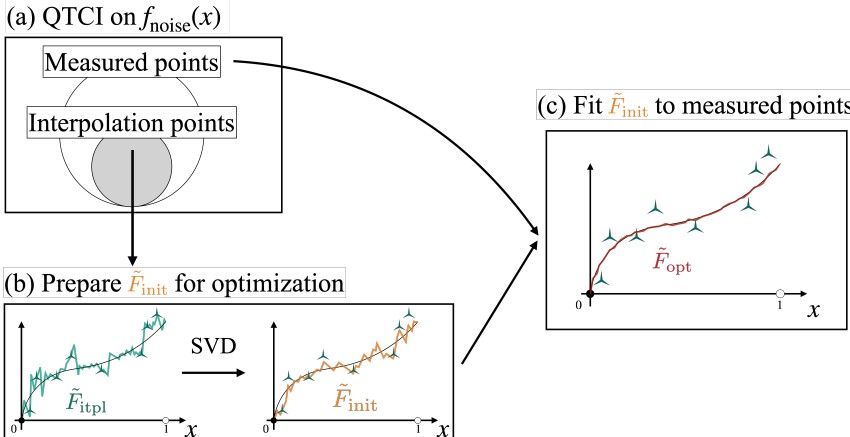

Figure 1: The optimization of QTTs proposed in this study. In Step (a), we apply the QTCI to the function $f_{\text{noise}}(\boldsymbol{x})$ evaluated with random noise and store measured points obtained from QTCI. In Step (b), we build $\tilde{F}_{\text{itpl}}$, which is composed of interpolation points obtained from QTCI, where the triangle marks in (b) represent the interpolation points. This is then compressed by applying SVD to obtain an initial guess of QTT, $\tilde{F}_{\text{init}}$. In Step (c), we optimize the $\tilde{F}_{\text{init}}$ by fitting it to the measured points obtained in Step (a) by applying the least squares method to reduce the effect of noise in $\tilde{F}_{\text{init}}$. This results in $\tilde{F}_{\text{opt}}$ that approximate the noise-free function $f(\boldsymbol{x})$.

rank in terms of the Frobenius norm, using a given error tolerance $\epsilon_{\text{SVD}}$. This tolerance is defined by

$$\epsilon_{\text{SVD}} = \frac{|\tilde{F}_{\text{TT}} - \tilde{F}'_{\text{TT}}|_{\text{F}}^2}{|\tilde{F}_{\text{TT}}|_{\text{F}}^2}, \tag{12}$$

where $|\cdots|_{\text{F}}^2$ indicates the Frobenius norm, $\tilde{F}_{\text{TT}}$ is the TT obtained from TCI and $\tilde{F}'_{\text{TT}}$ is the other TT after SVD. In this paper, we perform SVD by setting the maximum bond dimension of $\tilde{F}'_{\text{TT}}$.

# 4 Adaptive sampling-based optimization of QTT for noisy functions

We propose a method for optimizing TTs for functions affected by random noise in function evaluations. Here, we optimize an initial guess of TT by fitting it to measured points obtained from TCI [12,16–18], which capture the global structure of the target functions. This optimization yields the optimized TT that accurately approximates the noise-free function. Especially for functions with continuous variables, we use the QTCI [21] in our proposed methods. We demonstrate our approach with a simple sine function example. More challenging examples, such as cases where the bond dimension is not exactly known, will be introduced in the next section.

## 4.1 Outline

Step (a)   Perform QTCI to learn $\tilde{F}_{\text{itpl}}$ from a function $f_{\text{noise}}(\boldsymbol{x})$ that is subject to random noise during evaluation, setting the maximum bond dimension of QTCI to $\tilde{\chi}$. At the same time, we store all the results of function evaluations on the measured points obtained from QTCI shown in Fig. 1(a).

Step (b)   Prepare the initial guess $\tilde{F}_{\mathrm{init}}$ for optimization by compressing $\tilde{F}_{\mathrm{itpl}}$ from maximum bond dimension $\tilde{\chi}$ to $\chi$ ($\chi \leq \tilde{\chi}$) using SVD shown in Fig. 1(b).

Step (c)   Starting from the initial guess $\tilde{F}_{\mathrm{init}}$ obtained in Step (b), we fit this QTT to the measured points using the non-linear least-squares method [26, 27] shown in Fig. 1(c). As a result, we get an optimized QTT, $\tilde{F}_{\mathrm{opt}}$ that approximate the noise-free function $f(\boldsymbol{x})$.

Each of these steps is explained in more detail below.

First, in Step (a), we construct the QTT representation $\tilde{F}_{\mathrm{itpl}}$ of the multivariable function $f_{\mathrm{noise}}(\boldsymbol{x})$ by applying the QTCI algorithm with a specified maximum bond dimension $\tilde{\chi}$. As explained in Section 3.1, the QTCI algorithm performs multiple function evaluations during its interpolation process. In this process, it records every evaluated index along with its corresponding function value, collectively referred to as the measured points. The total number of these recorded measured points is denoted by $N^{\mathrm{TCI}}$. Among these measured points, QTCI adaptively selects a subset of indices for use in the interpolation. We refer to these as the interpolation points. Figure 1 (a) illustrates the relationship between the measured points and the interpolation points.

Note that $\tilde{F}_{\mathrm{itpl}}$ strictly interpolates the function values, which may contain noise, at the interpolation points chosen adaptively by the QTCI algorithm. Although we do not provide a rigorous theoretical justification that applying TCI to a noisy function will recover the global structure of the underlying noise-free function, we nevertheless expect that both the resulting QTT $\tilde{F}_{\mathrm{itpl}}$ and these measured points naturally capture the global structure of the original noise-free function $f(\boldsymbol{x})$. This expectation is supported by the observation that, when the QTCI algorithm reaches its prescribed maximum bond dimension $\tilde{\chi}$, the essential structural features of the function are likely to be captured within the interpolation error of TCI, tolerance. We then exploit the measured points for fitting in the subsequent optimization step. Note that $\tilde{\chi}$ is treated as a hyperparameter in the QTCI procedure.

In Step (b), we employ SVD to reduce the maximum bond dimension of $\tilde{F}_{\mathrm{itpl}}$ from $\tilde{\chi}$ to a smaller value $\chi$, which is treated as a hyperparameter. This truncation process yields a new QTT, denoted as $\tilde{F}_{\mathrm{init}}$, which serves as the initial guess for the subsequent fitting process. By reducing the bond dimension, we decrease the number of optimization parameters, which helps to improve the efficiency and stability of the optimization. Moreover, since the larger bond dimension $\tilde{\chi}$ may capture noise and lead to overfitting, we expect that a more compact representation with bond dimension $\chi$ is sufficient to describe the underlying noise-free function. The procedure for setting the hyperparameters $\chi$ and $\tilde{\chi}$ is discussed in Sec. 5.2.3.

In Step (c), starting from $\tilde{F}_{\mathrm{init}}$ obtained in Step (b), we optimize elements of $\tilde{F}_{\mathrm{init}}$ (i.e., its variational parameters $\boldsymbol{\theta}$) to reduce the effect of noise in this initial tensor. Specifically, we perform a gradient-based nonlinear least-squares minimization of the following cost function [26, 27]. Thus, all elements of the $\tilde{F}_{\mathrm{init}}$, $\boldsymbol{\theta}$ are optimized simultaneously.

$$\mathrm{cost}(\boldsymbol{\theta}) \equiv \sum_{i=1}^{N^{\mathrm{TCI}}} \left| z_i - \tilde{F}_{\mathrm{init}}(\boldsymbol{\sigma}_i) \right|^2 , \tag{13}$$

where $z_i$ ($i = 1, \cdots, N^{\mathrm{TCI}}$) denotes the value of $f_{\mathrm{noise}}$ at the measured point $\boldsymbol{\sigma}_i$. $\tilde{F}_{\mathrm{init}}(\boldsymbol{\sigma}_i)$ is the value of $\tilde{F}_{\mathrm{init}}$ at the $\boldsymbol{\sigma}_i$. The number of variational parameters corresponds to the total number of elements in the QTT representation of $\tilde{F}_{\mathrm{init}}$, is of order $\mathcal{O}(\mathcal{L}\chi^2)$, where $\mathcal{L}$ is the number of tensors and $\chi$ is the maximum bond dimension after truncation. We denote the number of optimizations performed during fitting as $n_{\mathrm{itr}}$. The application of the proposed method is presented in Sec. 5. For details on the bond dimensions and specific parameter settings used in the application example, refer to Sec. 5.2.3 and 5.2.4, respectively. After optimization, we get the optimized TT that accurately approximates the noise-free function.

## 4.2 Demonstration

As a simple demonstration, we consider the sine function $f(x) = \sin(2\pi x)$ defined over the interval $x \in [0, 1)$. We add Gaussian noise in the form of a normal distribution $N(\mu, \sigma^2)$ with mean $\mu$ and standard deviation $\sigma$, resulting in the function $f_{\text{noise}}(x, \mu, \sigma) = f(x) \times (1 + N(\mu, \sigma^2))$.

The numerical details are as follows. We set $\mathcal{R} = 12$, the mean of the normal distribution $\mu = 0$, $\tilde{\chi} = 6$, and $\chi = 2$. The choice of $\chi = 2$ is based on the exact bond dimension required for representing $f(x) = \sin(2\pi x)$ in QTT, while $\tilde{\chi} = 6$ is selected to larger value in order to increase the number of fitting points which capture the global structure of the target function. To determine $\tilde{\chi}$, we gradually increased the bond dimension and monitored the TCI error, as described in Section 3.1. We then selected the bond dimension at which the TCI error decreased and began to plateau. For reference, the average TCI error over 20 runs was approximately 0.351 for $\sigma = 0.1$ and 0.0381 for $\sigma = 0.01$, respectively. This choice is based on the assumption that a decreasing TCI error indicates successful extraction of the global structure of the noise-free target function from the noisy function.

Also, we set the number of fitting iterations $n_{\text{itr}} = 500$. This number was set after observing the decrease in the cost function during iterations, ensuring that the cost function converged sufficiently and remained nearly constant. We repeated this procedure 20 times using different realizations of noise.

Before presenting the results of our complete method, we first evaluate the impact of Step (b) by comparing the results obtained with and without bond-dimension compression. Figures 2(a) and (b) show the results for $\sigma = 0.1$ and $\sigma = 0.01$, respectively. Here, $\tilde{F}_{\text{opt}}$ refers to the optimized QTT obtained when Step (b) is applied (i.e., the bond dimension is compressed to $\chi = 2$ before optimization), whereas $\tilde{\tilde{F}}_{\text{opt}}$ (no compression) refers to the QTT optimized directly from the initial tensor with $\tilde{\chi} = 6$, skipping Step (b). The results demonstrate that applying Step (b) leads to improved accuracy, thereby validating the role of bond-dimension compression in suppressing noise and avoiding overfitting, as discussed in Sec. 4.1.

Fig. 3(a) shows results for $\sigma = 0.1$. Our proposed method not only brings the mean values closer to those of the noise-free function but also reduces the variance. Furthermore, the results for absolute error demonstrate that the proposed method reduces the average effect of noise on the function in terms of absolute error.

Fig. 3(b) shows results for smaller $\sigma = 0.01$. One can see that, even with a further reduction of the noise effect, both variance and absolute error have been reduced by the proposed method.

These results demonstrate the effectiveness of our proposed method. By applying Step (a), we are able to capture the global structure of the underlying noise-free function from the noisy function through QTCI. Step (b) not only reduces the number of variational parameters used in the optimization but also helps suppress the influence of noise slightly. Finally, Step (c) further reduces the effect of noise and enables the construction of a QTT representation that is closer to the true, noise-free function.

# 5 Application to quantum simulation based on pseudo-imaginary-time evolution

## 5.1 Setup

We use our proposed method to learn QTTs of two-time correlation functions, calculated on a quantum simulator, where noise such as Trotter error and shot noise are included. We then

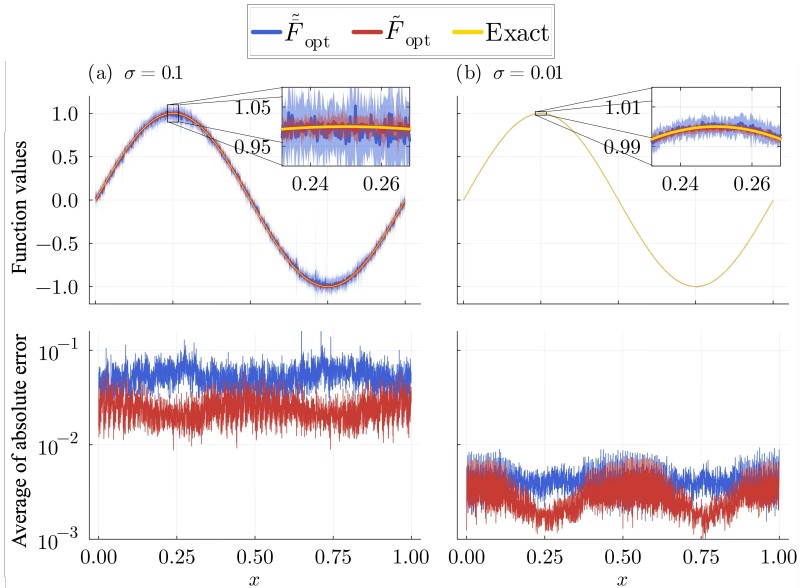

Figure 2: Results for our method applied to $f_{\text{noise}}(x,\mu,\sigma) = f(x) \times (1 + N(\mu,\sigma^2))$ with standard deviations $\sigma = 0.1$ and $\sigma = 0.01$ are shown in (a) and (b), respectively. In each figure, the upper panel shows the mean and variance across 20 runs, and the lower panel shows the mean of the absolute error. The insets within the upper panels provide magnified views of the region around $x = 0.25$. Here, $\tilde{F}_{\text{opt}}$ is the optimized QTT obtained after applying Step (b) (i.e., compressing the bond dimension to $\chi = 2$ before optimization), $\tilde{\tilde{F}}_{\text{opt}}$ is the QTT optimized directly from the initial tensor with $\tilde{\chi} = 6$, skipping Step (b), and "Exact" denotes the noise-free $f(x)$.

apply these optimized QTTs to quantum simulation based on pseudo-imaginary-time evolution to robustly compute the ground-state energy. For details on this pseudo-imaginary-time evolution method, please refer to Ref. [28].

### 5.1.1 Imaginary-time evolution

Using the traceless Hamiltonian $\bar{H}$ (where $\text{Tr}(\bar{H}) = 0$) and a parameter $E_0$, we redefine the Hamiltonian as $H = \bar{H} - E_0\mathbb{I}$. Let $E_g$ be the ground-state energy of $\bar{H}$ obtained through exact diagonalization. The parameter $E_0$ is a hyperparameter necessary for determining the ground-state energy. Now, let $\beta$ be the imaginary-time, $|\Psi(0)\rangle$ the initial state, and $O$ a physical quantity. In the imaginary-time evolution method, the expectation value of the physical quantity $O$ at time $\beta$, denoted as $\langle O \rangle (\beta)$, is calculated as:

$$\langle O \rangle (\beta) = \frac{\langle \Psi(0)| e^{-\beta H} O e^{-\beta H} |\Psi(0)\rangle}{\langle \Psi(0)| e^{-2\beta H} |\Psi(0)\rangle} \equiv \frac{\langle O \rangle (-i\beta, i\beta)}{\langle 1 \rangle (-i\beta, i\beta)}, \tag{14}$$

where 1 represents the identity operator. In particular, if $O = \bar{H}$, the energy expectation value $\langle \bar{H} \rangle (\beta)$, calculated via the imaginary-time evolution method, is known to converge to the ground-state energy $E_g$ of $\bar{H}$ as $\beta$ approaches infinity:

$$\lim_{\beta \to \infty} \langle \bar{H} \rangle (\beta) = E_g . \tag{15}$$

However, as the size of the system increases, the size of $\bar{H}$ and $H$ when represented as matrices also grow exponentially, making the calculations in Eqs. (14) and (15) difficult. To perform this imaginary-time evolution, several quantum algorithms have been proposed to approximate it using unitary operators, enabling its implementation on quantum computers [28–31].

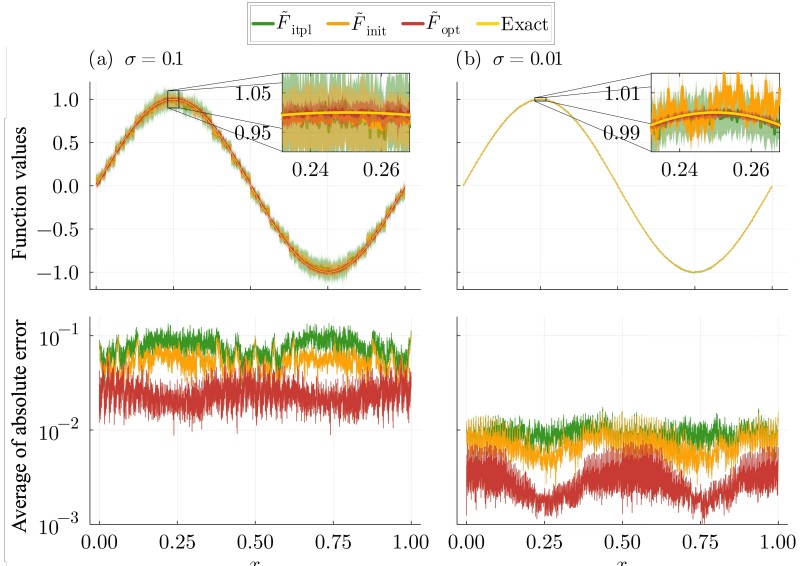

Figure 3: The setup and plotting same as in Fig. 2, but here we show results for our complete proposed method, including all three steps (a)–(c). $\tilde{F}_{\text{itpl}}$ is the QTT from QTCI, $\tilde{F}_{\text{init}}$ the QTT with the bond dimension compressed prior to optimization, $\tilde{F}_{\text{opt}}$ the optimized QTT, and "Exact" the noise-free $f(x)$.

### 5.1.2 Pseudo-imaginary-time evolution

We review the formalism based on Ref. [28]. We approximate the imaginary-time evolution operator $e^{-\beta H}$ by using the real-time evolution operator $e^{-iHt}$ as follows:

$$e^{-\beta H} \approx G(H) = \int_{-\infty}^{\infty} dt\, g(t)\, e^{-iHt}\,, \tag{16}$$

where

$$g(t) = f_{\text{L}}(\beta, t) \cdot f_{\text{G}}(1, 0, \tau, \sqrt{\beta^2 + t^2}) = \frac{1}{\pi} \frac{\beta}{\beta^2 + t^2} e^{-\frac{\beta^2 + t^2}{2\tau^2}}\,, \tag{17}$$

where $g(t)$ is the product of Lorentz and Gaussian functions. The Lorentz function is $f_{\text{L}}(\beta, t) = \frac{1}{\pi} \frac{\beta}{\beta^2 + t^2}$, and the Gaussian function is $f_{\text{G}}(A, \mu, \sigma, x) = A \exp\left(-\frac{(x-\mu)^2}{2\tau^2}\right)$. The parameter $\tau$ corresponds to the standard deviation of the Gaussian function. In the following, we refer to this method as pseudo-imaginary-time evolution.

Using the complementary error function, $\text{erfc}(x) = \frac{2}{\sqrt{\pi}} \int_x^{\infty} e^{-t^2} dt$ and the eigenvalues $\omega$ of $H$, Eq. (16) can be written as

$$G(\omega) = \sum_{\eta=\pm} \frac{1}{2} e^{\eta\beta\omega} \text{erfc}\left(\frac{\beta + \eta\omega\tau^2}{\sqrt{2}\tau}\right). \tag{18}$$

Here, the error in Eq. (16) is bounded using the matrix2-norm ($\|\cdot\|_2$) when $H$ is positive semidefinite, i.e., when $\Delta E = E_g - E_0 \geq \frac{\beta}{\tau^2}$, as follows:

$$\|G(H) - e^{-\beta H}\|_2 \leq \gamma_G = e^{-\frac{\Delta E^2 \tau^2}{2}}\,. \tag{19}$$

Next, the expectation value of the physical quantity $O$, $\langle O \rangle(\beta)$, can be approximated by replacing the imaginary-time evolution $e^{-\beta H}$ with Eq. (16) and thus translated into an integral calculation using the real-time evolution operator:

$$\langle O \rangle_G(\beta) = \frac{\langle O \rangle_G(-i\beta, i\beta)}{\langle 1 \rangle_G(-i\beta, i\beta)}\,, \tag{20}$$

$$\langle O \rangle_G (-i\beta, i\beta) = \langle \Psi(0)| G(H) O G(H) |\Psi(0)\rangle$$
$$= \int dt dt' g(t) g(t') \langle O \rangle (t, t'), \qquad (21)$$

$$\langle O \rangle (t, t') = \langle \Psi(0)| e^{iHt'} O e^{-iHt} |\Psi(0)\rangle$$
$$= e^{iE_0(t-t')} \langle \Psi(0)| e^{i\bar{H}t'} O e^{-i\bar{H}t} |\Psi(0)\rangle$$
$$= e^{iE_0(t-t')} \overline{\langle O \rangle}(t, t'), \qquad (22)$$

where $\langle \Psi(0)| e^{i\bar{H}t'} O e^{-i\bar{H}t} |\Psi(0)\rangle$ is denoted as $\overline{\langle O \rangle}(t, t')$. Next, we consider replacing the infinite integration range with a finite one $T$ in Eq. (16) as

$$G_T(H) = \int_{-T}^{T} dt g(t) e^{-iHt}. \qquad (23)$$

The error between $G(H)$ and $G_T(H)$ is given by

$$\|G_T(H) - G(H)\|_2 \leq \gamma_T = \frac{\sqrt{2}\tau}{\sqrt{\pi}\beta} e^{-\frac{T^2}{2\tau^2}}. \qquad (24)$$

Using Eq. (23), the finite integration range versions of Eqs. (20) and (21) are defined as follows:

$$\langle O \rangle_{G_T}(\beta) = \frac{\langle O \rangle_{G_T}(-i\beta, i\beta)}{\langle 1 \rangle_{G_T}(-i\beta, i\beta)}, \qquad (25)$$

$$\langle O \rangle_{G_T}(-i\beta, i\beta) = \langle \Psi(0)| G_T(H) O G_T(H) |\Psi(0)\rangle$$
$$= \int_{-T}^{T} dt dt' g(t) g(t') \langle O \rangle (t, t'), \qquad (26)$$

where the integration range $T$ and the constant $E_0$ are crucial parameters related to the accuracy of the expectation value $\langle O \rangle(\beta)$. The optimal selection of these parameters is beyond the scope of this study. In this research, the parameters are fixed as stated in Sec. 5.2.4, and their sufficiency for achieving adequate accuracy is verified.

We consider calculating the ground-state energy $\langle \bar{H} \rangle(\beta)$ of a given Hamiltonian $\bar{H}$ (where $O = \bar{H}$):

$$E_g \sim \langle \bar{H} \rangle(\beta) \simeq \langle \bar{H} \rangle_{G_T}(\beta) = \frac{\langle \bar{H} \rangle_{G_T}(-i\beta, i\beta)}{\langle 1 \rangle_{G_T}(-i\beta, i\beta)}. \qquad (27)$$

### 5.1.3 Monte Carlo method

This section describes a simplified version of the quantum algorithm from [28] for calculating the expectation value $\langle O \rangle_{G_T}(-i\beta, i\beta)$ in Eq. (26) using the Monte Carlo method. The method involves independently sampling times $t$ and $t'$ from probability density functions $g(t)$ and $g(t')$ within $[-T, T]$, and evaluating the expectation value by measuring the two-point real-time correlation function $\langle O \rangle(t, t')$ at the sampled times on a quantum simulator. The expectation value is given by:

$$\langle O \rangle_{G_T}(-i\beta, i\beta) = C^2 \int_{-T}^{T} dt dt' P(t, t') \langle O \rangle(t, t') \sim \frac{C^2}{N^{\text{MC}}} \sum_{i=1}^{N^{\text{MC}}} \langle O \rangle(t_i, t_i'), \qquad (28)$$

where $C = \int_{-T}^{T} dt g(t) = \int_{-T}^{T} dt' g(t')$ is the normalization constant, and $P(t, t') = \frac{g(t)g(t')}{C^2}$ is the product of the independent probability density functions. The Monte Carlo method converges to the desired expectation value at a rate of $\mathcal{O}\left(\frac{1}{\sqrt{N^{\text{MC}}}}\right)$ with respect to the number of samples $N^{\text{MC}}$.

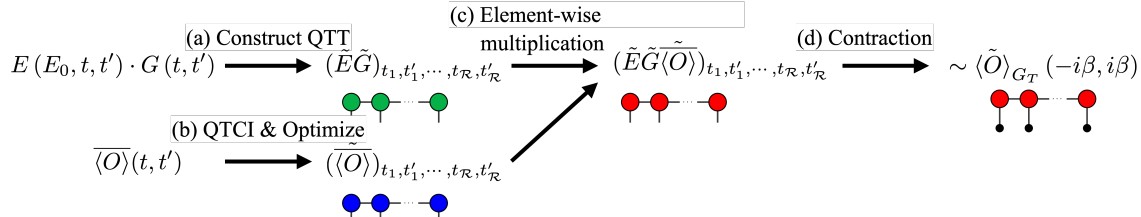

Figure 4: Calculating $\langle\tilde{O}\rangle_{G_T}(-i\beta, i\beta)$ using our proposed method through following Steps. First, in Step (a), we construct $(\tilde{E}\tilde{G})_{t_1,t_1',...,t_\mathcal{R},t_\mathcal{R}'}$ using the QTCI method. Next, in Step (b), we apply our proposed method to $\overline{\langle O\rangle}(t,t')$. Then, in Step (c), we perform element-wise multiplication of the QTTs obtained in Steps (a) and (b), resulting in $(\tilde{E}\tilde{G}\overline{\langle O\rangle})_{t_1,t_1',...,t_\mathcal{R},t_\mathcal{R}'}$. Finally, in Step (d), we integrate $(\tilde{E}\tilde{G}\overline{\langle O\rangle})_{t_1,t_1',...,t_\mathcal{R},t_\mathcal{R}'}$ over the range of $[-T, T)$ to compute $\langle\tilde{O}\rangle_{G_T}(-i\beta, i\beta)$.

### 5.1.4 Tensor network-based method

The $\langle O\rangle_{G_T}(-i\beta, i\beta)$ can be directly calculated using QTT as follows:

$$
\begin{aligned}
\langle O\rangle_{G_T}(-i\beta, i\beta) &= \int_{-T}^{T} dt\,dt' e^{iE_0(t-t')} g(t) g(t') \langle\Psi(0)| e^{i\tilde{H}t'} O e^{-i\tilde{H}t} |\Psi(0)\rangle \\
&= \int_{-T}^{T} dt\,dt' E(E_0,t,t') G(t,t') \overline{\langle O\rangle}(t,t') \\
&\simeq \langle\tilde{O}\rangle_{G_T}(-i\beta, i\beta),
\end{aligned}
\tag{29}
$$

where $E(E_0,t,t') = e^{iE_0(t-t')}$, $G(t,t') = g(t)g(t')$, and $\overline{\langle O\rangle}(t,t') = \langle\Psi(0)| e^{i\tilde{H}t'} O e^{-i\tilde{H}t} |\Psi(0)\rangle$. Each QTT is represented as $\tilde{E}_{t_1,t_1',...,t_\mathcal{R},t_\mathcal{R}'}$, $\tilde{G}_{t_1,t_1',...,t_\mathcal{R},t_\mathcal{R}'}$, and $\overline{\langle O\rangle}_{t_1,t_1',...,t_\mathcal{R},t_\mathcal{R}'}$, respectively. Using QTTs, the calculation of $\langle\tilde{O}\rangle_{G_T}(-i\beta, i\beta)$ that approximate $\langle O\rangle_{G_T}(-i\beta, i\beta)$ involves the following four Steps for a given value of $E_0$ (see Fig. 4):

Step (a)  Learn $\tilde{G}_{t_1,t_1',...,t_\mathcal{R},t_\mathcal{R}'}$ using QTCI. Then, we perform element-wise multiplication of $\tilde{E}_{t_1,t_1',...,t_\mathcal{R},t_\mathcal{R}'}$ with exact bond dimension and $\tilde{G}_{t_1,t_1',...,t_\mathcal{R},t_\mathcal{R}'}$ to construct the $(\tilde{E}\tilde{G})_{t_1,t_1',...,t_\mathcal{R},t_\mathcal{R}'}$. Details of the element-wise multiplication of QTTs are described in Appendix A.

Step (b)  Learn $\overline{\langle O\rangle}_{t_1,t_1',...,t_\mathcal{R},t_\mathcal{R}'}$ using our proposed method. Here, each element of the correlation function is computed using a quantum simulator, which introduces errors such as Trotter error and shot noise.

Step (c)  Perform element-wise multiplication of $(\tilde{E}\tilde{G})_{t_1,t_1',...,t_\mathcal{R},t_\mathcal{R}'}$ and $\overline{\langle O\rangle}_{t_1,t_1',...,t_\mathcal{R},t_\mathcal{R}'}$, which are obtained in Steps (a) and (b) respectively. As a result, we obtain $(\tilde{E}\tilde{G}\overline{\langle O\rangle})_{t_1,t_1',...,t_\mathcal{R},t_\mathcal{R}'}$ that approximate the $E(E_0,t,t')\cdot G(t,t')\cdot\overline{\langle O\rangle}(t,t')$.

Step (d)  Perform integration over the range $[-T, T)$ for the $(\tilde{E}\tilde{G}\overline{\langle O\rangle})_{t_1,t_1',...,t_\mathcal{R},t_\mathcal{R}'}$ obtained in Step (c) to calculate the integral value of $\langle\tilde{O}\rangle_{G_T}(-i\beta, i\beta)$. For more information on how TTs and integration are related, please refer to Appendix A.

### 5.1.5 Ground-state energy calculations based on our proposed method

We repeat four Steps in Sec. 5.1.4 while varying the value of $E_0$ over the range $[\tilde{E}_g - E, \tilde{E}_g + E]$ with step size $N_{E_0}$ included in $\langle \tilde{O} \rangle_{G_T}(-i\beta, i\beta)$, the ground-state energy can be determined as the minimal value (see Algorithm 1). Here, $\tilde{E}_g$ is set close to the ground-state energy $E_g$ obtained through mean-field approximation or similar techniques. In this study, we set $\tilde{E}_g = E_g$ and $E = 2$ as shown in Table 1.

---

**Algorithm 1** Ground-state energy solver based on quantics tensor trains

---

**Require:** $\bar{H}, \tilde{E}_g, E, \beta, \tau, T, \mathcal{R}, \tau_{\text{TCI}}, \tilde{\chi}, \chi, n_{\text{itr}}, N_t, M_s$.

1: Prepare $\tilde{G}_{t_1, t'_1, \ldots, t_{\mathcal{R}}, t'_{\mathcal{R}}}$ using the QTCI according to $\mathcal{R}$ and $\tau_{\text{TCI}}$.

2: Learn $\langle \overline{\tilde{H}} \rangle_{t_1, t'_1, \ldots, t_{\mathcal{R}}, t'_{\mathcal{R}}}$ using our proposed method based on $\mathcal{R}, \tilde{\chi}, \chi, n_{\text{itr}}$.

3: Take element-wise multiplication of $\tilde{G}_{t_1, t'_1, \ldots, t_{\mathcal{R}}, t'_{\mathcal{R}}}$ and $\langle \overline{\tilde{H}} \rangle_{t_1, t'_1, \ldots, t_{\mathcal{R}}, t'_{\mathcal{R}}}$ to obtain $(\tilde{G} \langle \overline{\tilde{H}} \rangle)_{t_1, t'_1, \ldots, t_{\mathcal{R}}, t'_{\mathcal{R}}}$.

4: Learn $\langle \tilde{1} \rangle_{t_1, t'_1, \ldots, t_{\mathcal{R}}, t'_{\mathcal{R}}}$ using our proposed method based on $\mathcal{R}, \tilde{\chi}, \chi, n_{\text{itr}}$.

5: Take element-wise multiplication of $\tilde{G}_{t_1, t'_1, \ldots, t_{\mathcal{R}}, t'_{\mathcal{R}}}$ and $\langle \tilde{1} \rangle_{t_1, t'_1, \ldots, t_{\mathcal{R}}, t'_{\mathcal{R}}}$ to obtain $(\tilde{G} \langle \tilde{1} \rangle)_{t_1, t'_1, \ldots, t_{\mathcal{R}}, t'_{\mathcal{R}}}$.

6: **for** $E_0 = \tilde{E}g - E$ to $\tilde{E}g + E$ with step size $N_{E_0}$ **do**

7:    Build $\tilde{E}_{t_1, t'_1, \ldots, t_{\mathcal{R}}, t'_{\mathcal{R}}}$ with exact bond dimension.

8:    Take element-wise multiplication of $\tilde{E}_{t_1, t'_1, \ldots, t_{\mathcal{R}}, t'_{\mathcal{R}}}$ and $(\tilde{G} \langle \overline{\tilde{H}} \rangle)_{t_1, t'_1, \ldots, t_{\mathcal{R}}, t'_{\mathcal{R}}}$ to build $(\tilde{E}\tilde{G} \langle \overline{\tilde{H}} \rangle)_{t_1, t'_1, \ldots, t_{\mathcal{R}}, t'_{\mathcal{R}}}$ and store the integral value of $\langle \tilde{H} \rangle_{G_T}(E_0)$ obtained by integrating $(\tilde{E}\tilde{G} \langle \overline{\tilde{H}} \rangle)_{t_1, t'_1, \ldots, t_{\mathcal{R}}, t'_{\mathcal{R}}}$ over the range of $[-T, T]$.

9:    Take element-wise multiplication of $\tilde{E}_{t_1, t'_1, \ldots, t_{\mathcal{R}}, t'_{\mathcal{R}}}$ and $(\tilde{G} \langle \tilde{1} \rangle)_{t_1, t'_1, \ldots, t_{\mathcal{R}}, t'_{\mathcal{R}}}$ to build $(\tilde{E}\tilde{G} \langle \tilde{1} \rangle)_{t_1, t'_1, \ldots, t_{\mathcal{R}}, t'_{\mathcal{R}}}$ and store the integral value of $\langle \tilde{1} \rangle_{G_T}(E_0)$ obtained by integrating $(\tilde{E}\tilde{G} \langle \tilde{1} \rangle)_{t_1, t'_1, \ldots, t_{\mathcal{R}}, t'_{\mathcal{R}}}$ over the range of $[-T, T]$.

10: **end for**

11: Define $\hat{E}(E_0) = \langle \tilde{H} \rangle_{G_T}(E_0) / \langle \tilde{1} \rangle_{G_T}(E_0)$.

**Ensure:** $\min_{E_0} \hat{E}(E_0) \leftarrow$ estimated ground-state energy

---

## 5.2 Numerical details

### 5.2.1 Software and hardware

We used `TensorCrossInterpolation.jl` for TCI [18], and `ITensors.jl` [32] for tensor contractions and SVD. We used `Kyulacs.jl` [33], a julia wrapper for `Qulacs` [34]. For the fitting TTs in our proposed method, the automatic differentiation and LBFGS optimization algorithm was used via `Optim.jl` [35], and `Zygote.jl` [36]. In our calculations, we used AMD EPYC 7702P 64-Core Processor. The total computation time for the largest system in this simulation, with 6 sites, was about 19 hours on a single CPU core.

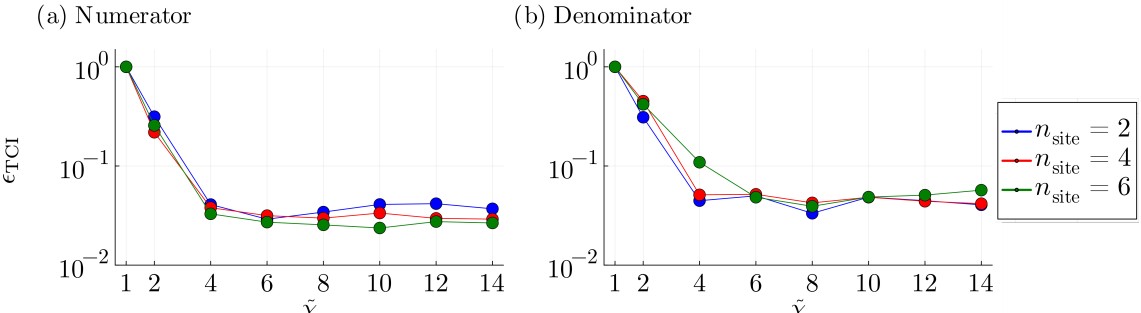

Figure 5: Normalized error $\epsilon_{\text{TCI}}$ as a function of the bond dimension $\tilde{\chi}$ for QTCI. (a) numerator $\langle \overline{H} \rangle (t, t')$, and (b) denominator $\langle \overline{1} \rangle (t, t')$. The parameters such as $\lambda$, $\beta$, $\tau$, $T$, $\mathcal{R}$, $N_{\text{t}}$, and $M_{\text{s}}$ are set as in Table 1.

### 5.2.2 Hamiltonian

For ground-state energy calculations, we consider the one-dimensional transverse-field Ising model Hamiltonian $\bar{H}$:

$$\bar{H} = -(2 - \lambda) \sum_{\langle i, j \rangle}^{n_{\text{site}}} \sigma_i^z \sigma_j^z - \lambda \sum_{i}^{n_{\text{site}}} \sigma_i^x , \tag{30}$$

where $\sigma^z$ and $\sigma^x$ are the Pauli-Z and Pauli-X operators, respectively. The model parameters include the number of sites $n_{\text{site}}$ and the strength of the transverse field $\lambda$.

### 5.2.3 Bond dimensions $\chi$ and $\tilde{\chi}$ in two-time correlation functions

Since the exact bond dimensions of the two-time correlation function are unknown, we apply QTCI on this function evaluated with noise to estimate the $\chi$ and $\tilde{\chi}$ roughly.

Figures 5(a) and (b) show interpolation errors estimated by QTCI normalized by the estimated absolute values of the functions, $\epsilon_{\text{TCI}}$. The error $\epsilon_{\text{TCI}}$ of QTCI initially decreases rapidly with increasing maximum bond dimension in the numerator and denominator, respectively. However, as $\tilde{\chi}$ increases, the error decrease stagnates, likely due to QTCI overfitting the noisy function evaluations, compromising approximation accuracy.

Based on this observation, we set $\tilde{\chi}$ to a value in the plateau region where $\epsilon_{\text{TCI}}$ saturates in the presence of noise. Specifically, we set $\tilde{\chi} = 4, 6, 10$ for $n_{\text{site}} = 2, 4, 6$, respectively. Next, we choose $\chi$ based on the fact that a noiseless function can be represented with a smaller bond dimension, whereas a function with noise requires a larger bond dimension. In particular, we set $\chi = 2, 4, 8$ for $n_{\text{site}} = 2, 4, 6$, respectively. After selecting these bond dimensions, we confirm that the precision of the ground-state energy is maintained. The values of $\chi$ are reasonable based on the results of the noise-free two-time function in Appendix C.

### 5.2.4 Other parameters

The parameters required to calculate the expectation value $\langle O \rangle_{G_T} (-i\beta, i\beta)$ are imaginary-time $\beta$, parameter $\tau$, integration range $T$, the number of bits $\mathcal{R}$, the tolerance of the QTCI $\tau_{\text{TCI}}$, the number of Trotter step $N_{\text{t}}$ and the number of measurement $M_{\text{s}}$ for computing the correlation function $\overline{\langle O \rangle}(t, t')$ on a quantum simulator and the number of the optimization iterations $n_{\text{itr}}$. At the stage of tuning the $E_0$ to calculate the ground-state energy, its range $E$ and step size $N_{E_0}$ are also needed. These parameters are uniformly applied regardless of the size $n_{\text{site}}$ as follows:

Table 1: Parameters used in the simulation.

| $\lambda$ | $\beta$ | $\tau$ | $T$ | $\mathcal{R}$ | $\tau_{\mathrm{TCI}}$ | $N_{\mathrm{t}}$ | $M_s$ | $n_{\mathrm{itr}}$ | $E$ | $N_{E_0}$ |
|---|---|---|---|---|---|---|---|---|---|---|
| 1.2 | 1.0 | 2.0 | 2.0 | 8 | $10^{-5}$ | 100 | 15000 | 500 | 2 | 40 |

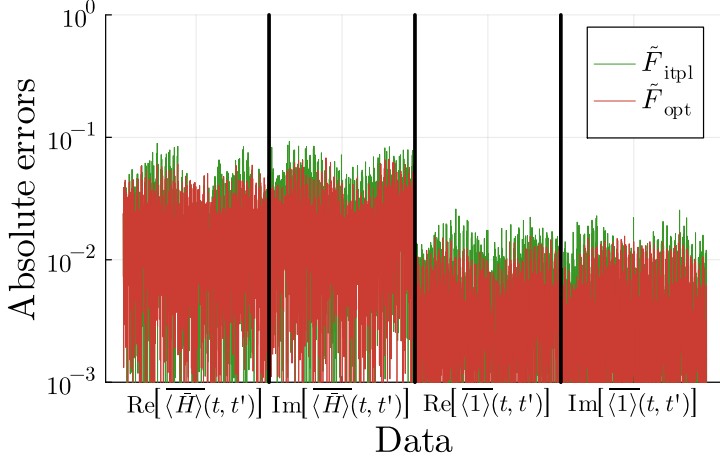

Figure 6: Results for the average absolute error between each of the values of the correlation function $\overline{\langle \tilde{\bar{H}} \rangle}\left(t, t'\right)$ and $\overline{\langle \tilde{1} \rangle}(t, t')$ obtained by applying the QTCI method (denoted as $\tilde{F}_{\mathrm{itpl}}$) or the proposed method (denoted as $\tilde{F}_{\mathrm{opt}}$) and the values from state vector simulation without noise. To enhance readability, we downsampled the displayed data to $1/50$ of the original resolution. The average absolute error was calculated over 20 trials, each using a different set of random numbers. The presented results include the one-dimensional reshaped data series of function values that depend on $t$ and $t'$. In both methods, for a system size of $n_{\mathrm{site}} = 4$, the parameters such as $\lambda$, $\beta$, $\tau$, $T$, $\mathcal{R}$, $N_{\mathrm{t}}$, and $M_s$ defined in Table 1 were used.

## 5.3 Demonstration

Figure 6 shows the results of learning $\overline{\langle \tilde{O} \rangle}_{t_1, t'_1, \ldots, t_{\mathcal{R}}, t'_{\mathcal{R}}}$ using the proposed method under the influence of noise, i.e., shot noise and Trotter errors. These errors arise while using a quantum simulator to evaluate each measured point of the two-time correlation function. We compare the results obtained using the QTCI method and the proposed method for estimating the correlation function $\overline{\langle O \rangle}(t, t')$ in QTT. This demonstrates that the proposed approach effectively mitigates the impact of noise, resulting in enhanced accuracy of two-time correlation functions.

Then, we move on to the results of the ground-state energy using quantum simulation based on pseudo-imaginary-time evolution. The parameters used for the calculations are detailed in Sec. 5.2.4. To ensure a fair comparison, the number of samples in the Monte Carlo method was set to match the number of evaluation points denoted as $N^{\mathrm{TCI}}$ obtained using the proposed method. The average number of evaluation points for the numerator and denominator were rounded up to the nearest decimal and set as $\bar{N}_n^{\mathrm{TCI}}$ and $\bar{N}_d^{\mathrm{TCI}}$, respectively. The Monte Carlo sampling numbers for the numerator and denominator were set as $N_n^{\mathrm{MC}} = \bar{N}_n^{\mathrm{TCI}}$ and $N_d^{\mathrm{MC}} = \bar{N}_d^{\mathrm{TCI}}$. For different system sizes, the pairs $\left(n_{\mathrm{site}}, \bar{N}_n^{\mathrm{TCI}}, \bar{N}_d^{\mathrm{TCI}}\right)$ were set as $(2, 767, 742)$, $(4, 1793, 1732)$, and $(6, 4480, 4592)$.

Figure 7(a) shows the relative error and its distribution of the ground-state energy calculated using three methods: the Monte Carlo method, the QTCI method, and the proposed method. These calculations were performed 20 times for each method and system sizes

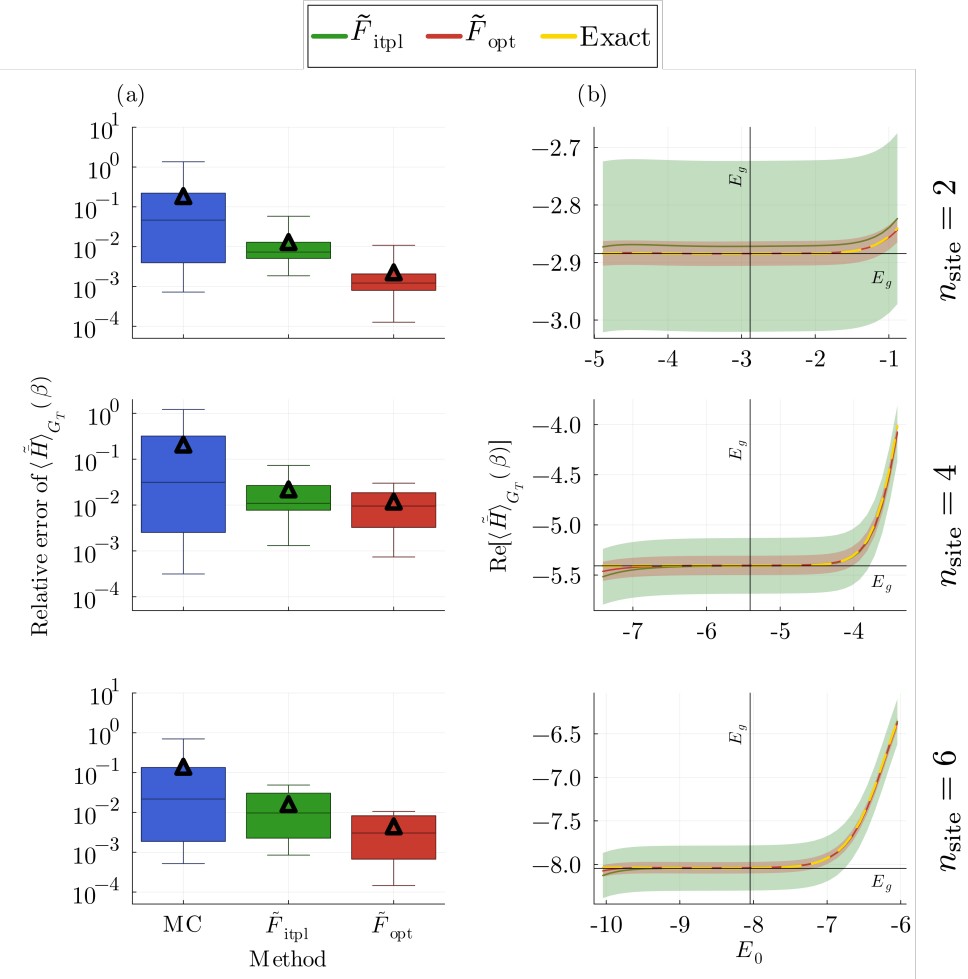

Figure 7: Results for the relative error of the ground-state energy computed by exact diagonalization method and each scheme (a) and $E_0$ dependency of the real part of the $\langle \tilde{\tilde{H}} \rangle_{G_T}(\beta)$ (b). Here, we compute its variation over 20 calculations using each scheme, with each different random seed. The triangle markers in (a) and the solid line in (b) represent the mean values, and the shading indicates the variance. The parameter set used for implementation is presented in Sec. 5.2.4. Due to the high variance of the Monte Carlo results, they are omitted from Fig. 7(b).

$n_{\text{site}} = 2, 4, 6$. When comparing the relative error in the expectation value of the Hamiltonian, the proposed method produces results that are nearer to the exact ground-state energy and show less variation than the other approaches.

Figure 7(b) compares the $E_0$ dependency of the real part of the $\langle \tilde{\tilde{H}} \rangle_{G_T}(\beta)$ calculated using the QTCI method and the proposed method. The proposed method outperforms QTCI in accurately and consistently determining the $E_0$ dependency under noisy conditions, using a comparable number of samples.

# 6 Conclusion

We propose a method to optimize a TT for a given function subject to random noise in their evaluations. In our method, we optimize an initial guess of a TT by fitting it to measured

points obtained from TCI. These measured points appropriately served as fitting points by capturing the overall structure of the target functions. As a result, we get the optimized TT that accurately approximates the noise-free function. In our experiments, we used QTCI in the proposed method and applied it to the sine function with added Gaussian noise and two-time correlation functions computed by a quantum simulator. Our approach resulted in a high-precision QTT compared to the QTCI method. Furthermore, we applied the optimized TTs of two-time correlation functions to quantum simulations based on pseudo-imaginary-time evolution for ground-state energy calculations. Our method achieved higher precision in determining the ground-state energy compared to the QTCI and Monte Carlo methods. Our study shows that the proposed method robustly learns TTs from functions under the influence of random noise in function evaluations.

We have several future directions. We aim to improve the accuracy of our proposed method by incorporating L1 regularization and smoothness conditions into the cost function during optimization. Additionally, combining the denoising techniques based on the Fourier transform proposed in [37] with our method may further enhance the denoising of QTT. Also, it may be possible to suppress the effect of noise in the correlation functions by incorporating the weight of $g(t)g(t')$ into the optimization process in the quantum simulation. We consider extending its application to compute imaginary-time Green's functions [38–40] based on pseudo-imaginary-time evolution, which poses challenges. Finally, this approach may also find other applications, such as quantum state tomography for photonic quantum computing [41].

## Acknowledgments

We are grateful to Marc K. Ritter and Jan von Delft for providing early access to the `TensorCrossInterpolation.jl` for TCI [18].

**Funding information**  R. S. was supported by the JSPS KAKENHI Grant No. 23KJ0295. H. S. was supported by JSPS KAKENHI Grants No. 21H01041, No. 21H01003, and No. 23H03817 as well as JST PRESTO Grant No. JPMJPR2012 and JST FOREST Grant No. JP-MJFR2232, Japan. This work was supported by Institute of Mathematics for Industry, Joint Usage/Research Center in Kyushu University. (FY2023 CATEGORY "Use of Julia in Mathematics and Physics" (2023a015). This work was also supported by the Center of Innovation for Sustainable Quantum AI (JST Grant Number JPMJPF2221).

## A  Element-wise multiplication and integration with tensor trains

### A.1  Element-wise multiplication

In this section, we consider computing $C(t)$ via the element-wise multiplication of functions $A(t)$ and $B(t)$:

$$C(t) = A(t) \cdot B(t). \tag{A.1}$$

In tensor network notation, this element-wise multiplication can be defined as:

$$C(t_1, t_2, \cdots, t_{\mathcal{R}}) = A(t_1, t_2, \cdots, t_{\mathcal{R}}) \cdot B(t_1, t_2, \cdots, t_{\mathcal{R}}). \tag{A.2}$$

This operation corresponds to specifying certain indices of tensor $C$ and assigning the corresponding element-wise multiplied values of tensors $A$ and $B$ to the indices in tensor $C$.

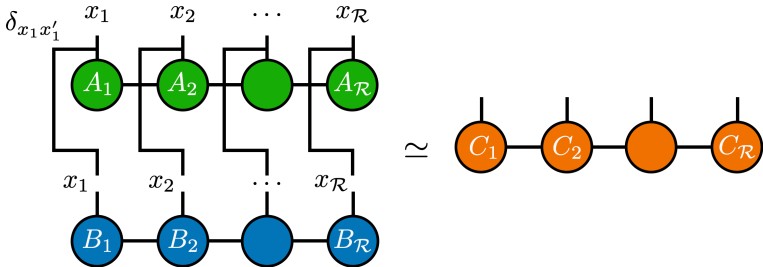

Figure 8: Diagram of the element-wise multiplication. The approximation means that the resulting TT is compressed by using SVD.

The functions $A(t)$, $B(t)$, and $C(t)$ can be represented in the TT. Specifically, $A(t)$ and $B(t)$ are expressed as $\mathcal{R}$-order TTs as follows:

$$A_{t_1 t_2 \cdots t_{\mathcal{R}}} \approx \sum_{\alpha_1=1}^{\chi_1^A} \cdots \sum_{\alpha_{\mathcal{R}-1}=1}^{\chi_{\mathcal{R}-1}^A} [A_1]_{1\alpha_1}^{t_1} \cdots [A_{\mathcal{R}}]_{\alpha_{\mathcal{R}-1}1}^{t_{\mathcal{R}}}, \tag{A.3}$$

$$B_{t_1 t_2 \cdots t_{\mathcal{R}}} \approx \sum_{\beta_1=1}^{\chi_1^B} \cdots \sum_{\beta_{\mathcal{R}-1}=1}^{\chi_{\mathcal{R}-1}^B} [B_1]_{1\beta_1}^{t_1} \cdots [B_{\mathcal{R}}]_{\beta_{\mathcal{R}-1}1}^{t_{\mathcal{R}}}. \tag{A.4}$$

To represent the element-wise multiplication $A(t) \cdot B(t)$ in MPO and TT, we use a delta function to express $A(t)$ as an MPO. The local bond indices $t_i$ and $t_i'$ with the same length scale are diagonalized using the delta function:

$$A_{t_1 t_2 \cdots t_{\mathcal{R}}}^{t_1' t_2' \cdots t_{\mathcal{R}}'} \approx \sum_{\alpha_1=1}^{\chi_1^A} \cdots \sum_{\alpha_{\mathcal{R}-1}=1}^{\chi_{\mathcal{R}-1}^A} [A_1]_{1\alpha_1}^{t_1 t_1'} \cdots [A_{\mathcal{R}}]_{\alpha_{\mathcal{R}-1}1}^{t_{\mathcal{R}} t_{\mathcal{R}}'}, \tag{A.5}$$

$$[A_k]_{\alpha_{j-1}\alpha_j}^{t_j t_j'} = [A_k]_{\alpha_{j-1}\alpha_j}^{t_j} \delta_{t_j, t_j'} \quad (k \in 1, \ldots, \mathcal{R}). \tag{A.6}$$

By expressing $A$ in the MPO representation, the element-wise multiplication $A(t) \cdot B(t)$ can be calculated through the contraction of MPO and TT. The element-wise multiplication diagram is shown in Fig. 8. If the bond dimensions of tensors $A$ and $B$ are denoted as $\chi_A$ and $\chi_B$, respectively, the computational complexity is $\mathcal{O}(\mathcal{R}\chi_A^2 \chi_B^2)$.

## A.2 Integration

We consider the QTT, $\tilde{F}$ of $\mathcal{L}$-order with a local dimension $d = 2$ and bond dimension $\chi$, of a function $f(x)$. The sum of the function $f(x)$ values at grid points with intervals of $\Delta x = 1/2^{\mathcal{L}}$ is equivalent to the Riemann sum, as follows:

$$\int_{x_0}^{x_{2^{\mathcal{L}}-1}} f(x)dx \approx \sum_{i=0}^{2^{\mathcal{L}}-1} f(x_i)\Delta x. \tag{A.7}$$

The element-wise sum operation, multiplied by $\Delta x$, is equivalent to contracting the QTT of $f(x)$ with the following equation:

$$I(\sigma_1, \ldots, \sigma_{\mathcal{L}}) = \Delta x \begin{bmatrix} 1 \\ 1 \end{bmatrix} \otimes \begin{bmatrix} 1 \\ 1 \end{bmatrix} \otimes \cdots \otimes \begin{bmatrix} 1 \\ 1 \end{bmatrix}. \tag{A.8}$$

The integration operation in QTT can be computed by contracting $\tilde{F}$ with tensors of bond dimension 1 filled with ones for each core tensor of the QTT. This can be extended to multivariate

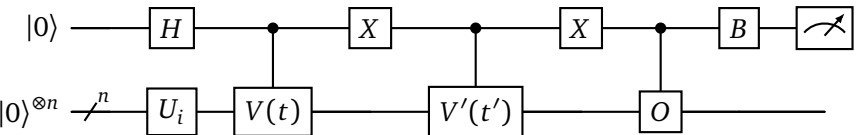

Figure 9: Quantum circuit for calculating $\overline{\langle O \rangle}(t, t')$. Here, we denote unitary operators $V(t) = e^{-i\bar{H}t}$, $V'(t') = e^{-i\bar{H}t'}$, and $U_i = H^{\otimes n}$, along with the unitary physical quantity $O$. Additionally, the $B$ gate is set to $B = H$ (Pauli-$X$ measurement) for calculating the real part of $\overline{\langle O \rangle}(t, t')$, and $B = HS^\dagger$ (Pauli-$Y$ measurement) for calculating the imaginary part.

functions. Diagrammatically, it can be represented as follows:

$$\sum_{\sigma_1, \ldots, \sigma_{\mathcal{L}}} \left\{ \sum_{i_1, \ldots, i_{\mathcal{L}-1}}^{\chi} [F_1]_{1i_1}^{\sigma_1} [F_2]_{i_1 i_2}^{\sigma_2} \cdots [F_{\mathcal{L}}]_{i_{\mathcal{L}-1}1}^{\sigma_{\mathcal{L}}} \right\} \Longleftrightarrow \quad \cdots \quad , \quad \bullet = \begin{pmatrix} 1 \\ 1 \\ \vdots \\ 1 \end{pmatrix} \right\} d. \;, \quad \text{(A.9)}$$

where the computational complexity is $\mathcal{O}(\mathcal{L}\chi^2)$.

## B Calculating the real-time correlation function $\langle O \rangle (t, t')$

The real-time correlation function $\langle O \rangle (t, t')$ can be rewritten as shown in Eq. (B.1):

$$
\begin{aligned}
\langle O \rangle (t, t') &= \langle \Psi(0) | e^{iHt'} O e^{-iHt} | \Psi(0) \rangle \\
&= e^{iE_0(t-t')} \langle \Psi(0) | e^{i\bar{H}t'} O e^{-i\bar{H}t} | \Psi(0) \rangle \\
&= e^{iE_0(t-t')} \overline{\langle O \rangle}(t, t').
\end{aligned}
\tag{B.1}
$$

The quantum circuit required for calculating $\overline{\langle O \rangle}(t, t')$ is shown in Fig. 9. Following the circuit in Fig. 9, the computation proceeds as follows:

$$
\begin{aligned}
|0\rangle \otimes |0\rangle^{\otimes n} &\xrightarrow{H \otimes U_i} \frac{1}{\sqrt{2}} (|0\rangle + |1\rangle) \otimes |\Psi(0)\rangle \\
&\xrightarrow{|0\rangle\langle 0| \otimes I^{\otimes n} + |1\rangle\langle 1| \otimes V(t)} \frac{1}{\sqrt{2}} \left\{ |0\rangle \otimes |\Psi(0)\rangle + |1\rangle \otimes V(t) |\Psi(0)\rangle \right\} \\
&\xrightarrow{X \otimes I^{\otimes n}} \frac{1}{\sqrt{2}} \left\{ |1\rangle \otimes |\Psi(0)\rangle + |0\rangle \otimes V(t) |\Psi(0)\rangle \right\} \\
&\xrightarrow{|0\rangle\langle 0| \otimes I^{\otimes n} + |1\rangle\langle 1| \otimes V'(t')} \frac{1}{\sqrt{2}} \left\{ |1\rangle \otimes V'(t') |\Psi(0)\rangle + |0\rangle \otimes V(t) |\Psi(0)\rangle \right\} \\
&\xrightarrow{X \otimes I^{\otimes n}} \frac{1}{\sqrt{2}} \left\{ |0\rangle \otimes V'(t') |\Psi(0)\rangle + |1\rangle \otimes V(t) |\Psi(0)\rangle \right\} \\
&\xrightarrow{|0\rangle\langle 0| \otimes I^{\otimes n} + |1\rangle\langle 1| \otimes O} \frac{1}{\sqrt{2}} \left\{ |0\rangle \otimes V'(t') |\Psi(0)\rangle + |1\rangle \otimes O V(t) |\Psi(0)\rangle \right\}.
\end{aligned}
\tag{B.2}
$$

To calculate the real part of $\overline{\langle O \rangle}(t, t')$, the $B$ gate is set to $H$ (Pauli-$X$ measurement). For the imaginary-time part, the gate is set to $HS^\dagger$ (Pauli-$Y$ measurement). For $\overline{\langle O \rangle}(t, t') = \langle \Psi(0) | V'^\dagger(t') O V(t) | \Psi(0) \rangle \equiv a + ib$ ($a, b \in \mathbb{R}$), we can proceed with the computation for Pauli-$X$ measurement and Pauli-$Y$ measurement case, respectively.

For the case of Pauli-$X$ measurement, the equation in Eq. (B.2) is transformed as follows:

$$\xrightarrow{H\otimes I^{\otimes n}} \frac{1}{\sqrt{2}}\left\{\frac{1}{\sqrt{2}}(|0\rangle+|1\rangle)\otimes V'(t')|\Psi(0)\rangle+\frac{1}{\sqrt{2}}(|0\rangle-|1\rangle)\otimes OV(t)|\Psi(0)\rangle\right\}$$
$$=\frac{1}{2}\Big[|0\rangle\otimes\big\{V'(t')|\Psi(0)\rangle+OV(t)|\Psi(0)\rangle\big\}+|1\rangle\otimes\big\{V'(t')|\Psi(0)\rangle-OV(t)|\Psi(0)\rangle\big\}\Big]. \tag{B.3}$$

The probability of measuring the first qubit as 0, denoted as $P_{x0}$, can be computed as follows:

$$\begin{aligned}
P_{x0} &= \left|\frac{V'(t')|\Psi(0)\rangle+OV(t)|\Psi(0)\rangle}{2}\right|^2 \\
&= \frac{1}{4}\big\{\langle\Psi(0)|V'^{\dagger}(t')+\langle\Psi(0)|V^{\dagger}(t)O^{\dagger}\big\}\big\{V'(t')|\Psi(0)\rangle+OV(t)|\Psi(0)\rangle\big\} \\
&= \frac{1+a}{2}.
\end{aligned} \tag{B.4}$$

The probability of measuring the first qubit as 1, denoted as $P_{x1}$, can be computed as follows:

$$\begin{aligned}
P_{x1} &= \left|\frac{V'(t')|\Psi(0)\rangle-OV(t)|\Psi(0)\rangle}{2}\right|^2 \\
&= \frac{1}{4}\big\{\langle\Psi(0)|V'^{\dagger}(t')-\langle\Psi(0)|V^{\dagger}(t)O^{\dagger}\big\}\big\{V'(t')|\Psi(0)\rangle-OV(t)|\Psi(0)\rangle\big\} \\
&= \frac{1-a}{2}.
\end{aligned} \tag{B.5}$$

Thus, the expectation value from the Pauli-X measurement is

$$\langle Z\rangle = (+1)\times P_{x0}+(-1)\times P_{x1} = a. \tag{B.6}$$

For the case of Pauli-$Y$ measurement, the equation in Eq. (B.2) is transformed as follows:

$$\xrightarrow{HS^{\dagger}\otimes I^{\otimes n}} \frac{1}{\sqrt{2}}\left\{\frac{1}{\sqrt{2}}(|0\rangle+|1\rangle)\otimes V'(t')|\Psi(0)\rangle-i\frac{1}{\sqrt{2}}(|0\rangle-|1\rangle)\otimes OV(t)|\Psi(0)\rangle\right\}$$
$$=\frac{1}{2}\Big[|0\rangle\otimes\big\{V'(t')|\Psi(0)\rangle-iOV(t)|\Psi(0)\rangle\big\}+|1\rangle\otimes\big\{V'(t')|\Psi(0)\rangle+iOV(t)|\Psi(0)\rangle\big\}\Big]. \tag{B.7}$$

The probability of measuring the first qubit as 0, denoted as $P_{y0}$, can be computed as follows:

$$\begin{aligned}
P_{y0} &= \left|\frac{V'(t')|\Psi(0)\rangle-iOV(t)|\Psi(0)\rangle}{2}\right|^2 \\
&= \frac{1}{4}\big\{\langle\Psi(0)|V'^{\dagger}(t')+i\langle\Psi(0)|V^{\dagger}(t)O^{\dagger}\big\}\big\{V'(t')|\Psi(0)\rangle-iOV(t)|\Psi(0)\rangle\big\} \\
&= \frac{1+b}{2}.
\end{aligned} \tag{B.8}$$

The probability of measuring the first qubit as 1, denoted as $P_{y1}$, can be computed as follows:

$$\begin{aligned}
P_{y1} &= \left|\frac{V'(t')|\Psi(0)\rangle+iOV(t)|\Psi(0)\rangle}{2}\right|^2 \\
&= \frac{1}{4}\big\{\langle\Psi(0)|V'^{\dagger}(t')-i\langle\Psi(0)|V^{\dagger}(t)O^{\dagger}\big\}\big\{V'(t')|\Psi(0)\rangle+iOV(t)|\Psi(0)\rangle\big\} \\
&= \frac{1-b}{2}.
\end{aligned} \tag{B.9}$$

Thus, the expectation value from the Pauli-Y measurement is:

$$\langle Z\rangle = (+1)\times P_{y0}+(-1)\times P_{y1} = b. \tag{B.10}$$

Thus, we can calculate the real-time correlation function $\langle O\rangle(t,t')$ by separating $\overline{\langle O\rangle}(t,t')$ into its real and imaginary parts. In this calculation, shot noise resulting from a finite number of measurements and Trotter errors due to the limited number of Trotter steps is included.

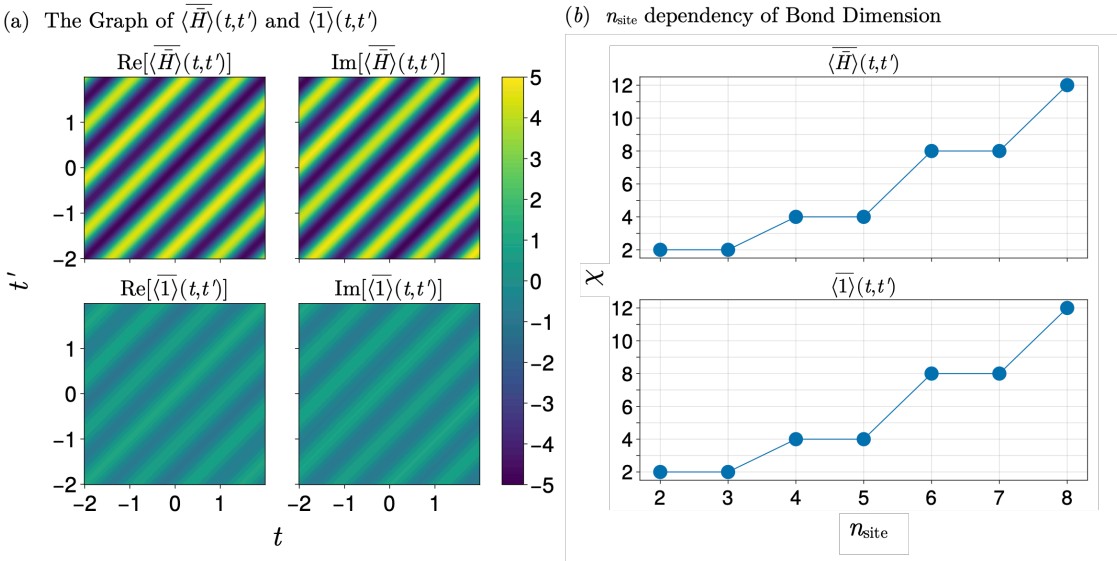

Figure 10: Computed correlation functions using state-vector simulation without shot noise, Trotter error (a), and their maximum bond dimensions (b). The QTT was constructed using SVD, setting its tolerance $10^{-10}$.

## C  Bond dimensions of two-time correlation functions without noise

We compute each element of the two-time correlation function using state vector simulation without noise, i.e., shot noise and Trotter error, and determine the bond dimension by compressing this correlation function using SVD.

Figure 10(a) shows the shape of the real-time correlation functions $\left\langle \bar{H} \right\rangle(t, t')$ and $\overline{\langle 1 \rangle}(t, t')$, calculated without shot noise and Trotter errors, over the interval $t, t' \in [-2, 2]$. The upper panel displays the real and imaginary parts of the correlation function $\left\langle \bar{H} \right\rangle(t, t')$ used for calculating the numerator of $\left\langle \bar{H} \right\rangle_{G_T}(\beta)$, while the lower panel shows those of the correlation function $\overline{\langle 1 \rangle}(t, t')$ used for the denominator. Here, the parameters are set to $\lambda = 1.2$, $n_{\text{site}} = 4$ and $\mathcal{R} = 8$. The two-time correlation functions exhibit a characteristic wave-like striped pattern along the 45-degree direction.

Figure 10(b) shows that the correlation function exhibits a low-rank structure. The maximum bond dimensions reach only around 12 for system sizes $n_{\text{site}}$ up to 8 sites.

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
