# Peer review of "Adaptive sampling-based optimization of quantics tensor trains for noisy functions: applications to quantum simulations"

_SciPost Physics, doi:SciPost Phys. 19, 038 (2025)_

## Round 1 · Referee Report · Anonymous (Referee 1) · 2025-1-2

Strengths
- The paper is very clearly written, and the main algorithm presented is straightforward to understand.
- The algorithm works well at reducing the errors, by about an order of magnitude, over the baseline approach to learning noisy functions with the TCI algorithm.
Weaknesses
Report
I think the paper should be accepted once the authors address the changes and questions below.
Requested changes
Requested changes: 1. in Section 3.1, the authors should explain or ideally give a citation for the statement that TCI is quasi-optimal in the maximum norm 2. at the beginning of page 10, there is a statement "where <\bar{O}>(t,t') is denoted as <\bar{O}>(t,t')". What is the meaning of this statement and is it a mistake? Otherwise it seems to have no content since it says something is equal to itself. 3. Figure 5 should be improved by not using a bar-chart format, unless the authors could explain a good reason. I think a line plot would be clearer. (Or is the chart actually a line that is very dense and ranges over t values?) 4. do the authors provide a reference or explanation for the algorithm to perform the least-squares minimization of a tensor train (step (c) on page 7)? If not, the authors should ideally provide a reference (perhaps from the MPS machine learning literature, such as Phys. Rev. X 8, 03101 if that seems relevant enough) or provide a short explanation of the steps in an Appendix.
Minor (optional) changes: 1. in the introduction to Section 4, the authors should write out "sine" instead of "sin. 2. at the end of the Acknowledgements, it says "IUse of Julia...". Should it be "Use of Julia..."?
Recommendation
Ask for minor revision

Kohtaroh Sakaue on 2025-03-15 [id 5292]
Thank you very much for your careful review of our manuscript and for your valuable comments and suggestions. We sincerely appreciate the time and effort you have put into evaluating our work. Your constructive feedback has been very helpful in improving the clarity and completeness of our paper. We address each of your comments below in detail.
We added a citation to [arXiv:2407.02454] in Section 3.1 to support the statement that TCI is quasi-optimal in the maximum norm. This reference provides a detailed analysis of the quasi-optimality properties of TCI and strengthens the theoretical foundation of our discussion.
We agree that the original statement was unclear and could be misleading. To clarify, we revised it to: "where \bra{\Psi\left(0\right)}e^{i\bar{H}t'}Oe^{-i\bar{H}t}\ket{\Psi\left(0\right)} is denoted as \overline{\expval{O}}(t,t')." This revision explicitly defines the notation and removes the redundant statement.
Initially, we used a line plot for Figure 5. However, due to the high density of the data points, the plot became difficult to interpret. To improve clarity, we downsampled the displayed data to 1/50 of the original resolution. Additionally, we adjusted the transparency of the $\tilde{F}_{\mathrm{opt}}$ plot to enhance readability. These modifications make the figure clearer while preserving the key information. Please refer to the revised Figure 5 attached for your review.
We added references to both [arXiv:1906.06329] and Phys. Rev. X 8, 031012 to support the least-squares minimization of a tensor train in step (c) on page 7. These references provide relevant background on the optimization techniques used in our approach and strengthen the theoretical foundation of our discussion.
Thank you for your careful attention to detail. We made the suggested corrections: "sin" has been changed to "sine" in the introduction to Section 4, and "IUse of Julia..." has been corrected to "Use of Julia..." in the Acknowledgements. We kindly ask you to verify these changes.
Attachment:
abs_err_1d_step_length_50.pdf

---

## Round 1 · Referee Report · Anonymous (Referee 2) · 2025-3-20

Strengths
- Propose a novel procedure which improves the TCI interpolation for fitting noisy functions
- Showcases benchmarks and applications of the method
Report
The authors propose a method to encode in an efficient way noisy functions in the tensor train (TT) format. Based on the benchmarks provided by the authors, the idea looks promising and I think it fits the Scipost requirements for publication.
While (with the exception of Step (c)) the authors explain the steps of the algorithm, what I find a bit missing is some discussion on why these steps work, especially for readers who have less familiarity on the TCI algorithm.
As an example, the authors state repeatedly that the measured points in the initial TCI interpolation* of Step (a) capture the global structure of the noise-free function f(x), something which seems reasonable but I fail to see how this can be rigorously justified.
- as a matter of fact, in the discussion in Sec. 4.1 the authors state "this measured point captures..." as if they were referring to a single point, I guess it's a typo, since both in the introduction as well as in the initial discussion in Sec4 the authors say that " the measured points capture.."
Regarding step (b), it's not clear to me what is the advantage of (over)fitting the function to a larger \tilde{\chi} and then subsequently compressing it, rather than just imposing a maximum \chi to the TCI directly in step (a). Is that in order to get more function evaluations for the later fitting done in step (c) ? In that case I imagine one could also pick some points on some regular grid, so I imagine there's another reason for that, I hope the authors can clarify this - In general, it would be interesting for me to compare what happens if one does overfitting+compression vs just fitting with smaller chi before performing step (c)
Finally, I agree with the other referee saying that the step (c) of the algorithm is not well explained, and I wish the authors could expand on it, since to my understanding it is a crucial part of the method. As a matter of fact, I'd be curious also to see how much does this final step reduce the error, compared to steps (a+b) only.
In summary, I think the method is definitely interesting but I'd like for the authors to discuss a bit more the intuition behind it, some more details on the last fitting and some basic benchmarking discussing the relevance of the intermediate step (b) vs (a) only with a restricted \chi and of the improvement given by the last fitting (c), even if only for the "basic" benchmark of sec 4.2
Requested changes
- extend discussion on the intuition behind the method (see my comments above)
- provide a more detailed description of step (c) of the procedure
-
discuss in a more detailed benchmark the relevance of steps (b) and (c), see my comments above
-
improve Fig5, which is not very clear (I don't understand if those are some kind of error bars, or heavily fluctuating lines, and whether the green bands are narrow or they are covered by the red ones)
Recommendation
Ask for minor revision
Thank you very much for your thoughtful and insightful review of our manuscript. We are grateful for your positive evaluation of our work and for highlighting both its strengths and areas where further clarification was needed. Your comments have prompted us to improve the exposition and provide additional explanations and benchmarks that we believe enhance the overall quality and accessibility of the paper. Below, we provide detailed responses to each of your comments.
- extend discussion on the intuition behind the method (see my comments above)
We thank the referee for pointing out the need to clarify the intuition behind the method, particularly in relation to Step (a) and Step (b) of our procedure.
Regarding Step (a): As explained in the manuscript, the QTCI algorithm adaptively selects interpolation points and builds a tensor-train representation $\tilde{F}_{\mathrm{itpl}}$ that strictly interpolates the noisy function values at those points. Despite the presence of noise, our empirical results suggest that both the chosen interpolation points and the resulting TT representation capture the main global features of the original, noise-free function $f(\boldsymbol{x})$. While we do not rigorously prove this in this work, we base our expectation on the empirical observation that, once QTCI algorithm reaches to its prescribed maximum bond dimension $\tilde{\chi}$, it allocates its degrees of freedom to the most structurally important directions in the data. At that stage, the interpolation error of TCI typically plateaus, indicating that the essential global features of the original function have been captured within the specified tolerance. We have revised the manuscript to clarify this reasoning for readers who are new to the TCI algorithm.
Regarding Step (b): The goal of Step (b) is to reduce the bond dimension from $\tilde{\chi}$ to a smaller value $\chi$ via TT compression using SVD. This step serves two purposes: (i) it reduces the number of variational parameters in the subsequent optimization in Step (c), thereby enhancing numerical stability and efficiency; and (ii) it mitigates overfitting effects that may have occurred in Step (a) due to the higher capacity of the tensor representation. Since setting to $\tilde{\chi}$ may partially capture noise, we expect that a more compact representation with reduced bond dimension $\chi$ can suppress such overfitting while still preserving the essential structure of the noise-free function. We have clarified this expectation and its role in the workflow in the revised manuscript as well.
We hope this additional discussion helps clarify the intuition behind the design of our method.
- provide a more detailed description of step (c) of the procedure
Thank you for pointing out the need for a more detailed description of Step~(c). Starting from the tensor $\tilde{F}_{\mathrm{init}}$ obtained in Step~(b), we fully optimize its variational parameters to reduce the effect of noise that may remain in this initial tensor. Concretely, we employ a gradient-based nonlinear least-squares minimization, in which the cost function is the sum of squared errors between the noisy function data and the values of the QTT $\tilde{F}_{\mathrm{init}}$ on the measured points. We optimize all elements of $\tilde{F}_{\mathrm{init}}$, so the total number of variational parameters of $\tilde{F}_{\mathrm{init}}$ scales as $\mathcal{O}(\mathcal{L} \chi^2)$, where $\mathcal{L}$ is the order of the tensor and $\chi$ is the maximum bond dimension after truncation in Step~(b). We denote the number of optimization iterations by $n_{\mathrm{itr}}$. The outcome of this optimization is an improved tensor, denoted $\tilde{F}_{\mathrm{opt}}$, that accurately approximates the underlying noise-free function. An example of this procedure appears in Sec. 4, with detailed bond-dimension and parameter settings in Sec. 4.2.
- discuss in a more detailed benchmark the relevance of steps (b) and (c), see my comments above
Thank you for your valuable comment.
In response to the comment for a more detailed benchmark, we compared the optimized QTT both with and without the bond-dimension compression of Step~(b). The case labeled $\tilde{F}_{\mathrm{opt}}$ corresponds to the scenario where the bond dimension is first compressed to $\chi=2$ before optimization, while $\tilde{\bar{F}}_{\mathrm{opt}}$ denotes the QTT optimized directly from the initial tensor with the bond dimension $\tilde{\chi}=6$ set in Step~(a), skipping Step~(b). As shown in the first figure in the attached file, performing Step~(b) leads to improved accuracy. This result supports our intuition that by reducing the bond dimension, we decrease the number of optimization parameters, which helps to improve the efficiency and stability of the optimization. Moreover, since a larger bond dimension $\tilde{\chi}$ may capture noise and cause overfitting, we expect that a more compact representation with a smaller bond dimension $\chi$ is sufficient to describe the underlying noise-free function.
- improve Fig5, which is not very clear (I don't understand if those are some kind of error bars, or heavily fluctuating lines, and whether the green bands are narrow or they are covered by the red ones)
Thank you for pointing this out. We agree that the original Figure 5 lacked clarity due to the high density of the data. In response, we have revised the figure to make it more interpretable: (1) we downsampled the data points by a factor of 50 to reduce visual clutter, and (2) we adjusted the transparency of the $\tilde{F}_{\mathrm{opt}}$ line so that overlapping regions between the two curves can be visually distinguished. We also confirm that the figure does not display error bars — the fluctuations visible in the original version were due to the pointwise variation in the absolute error values across $t$ and $t'$. We believe the updated version improves visibility and better communicates the comparison between the two methods. Please refer to the revised Figure 5 in the second figure in the attached file for your assessment .
Attachment:

---

## Round 2 · Referee Report · Anonymous (Referee 1) · 2025-6-13

Strengths

  1. The proposed algorithm works well in the cases shown and is explored in detail.
  2. The second version of the paper goes to greater lengths to explain the steps of the algorithm, provide references, and justify why the algorithm works.

Weaknesses

The algorithm may fail in more challenging cases not yet explored. Here I believe the authors' suggestion of making the last step of their algorithm include a regularization could help in such cases.

Report

The authors have addressed my requested changes and I think those of the other referee in a satisfactory way that has improved the paper. I already believed that the paper meets the acceptance criteria, conditional on the requested changes, so now the paper should be published.

Recommendation

Publish (meets expectations and criteria for this Journal)

---

## Round 2 · Referee Report · Anonymous (Referee 2) · 2025-6-17

Report

I thank the authors for addressing the points raised in the first report. I believe the manuscript has improved that it can be published in its current form.

Recommendation

Publish (meets expectations and criteria for this Journal)

---

## Round 2 · Author Response

Dear Editor and Referees,

We would like to sincerely thank the Editor and Referees for their time and effort in reviewing our manuscript and for providing thoughtful and constructive feedback. We are especially grateful for the positive assessments from both Referee 1 and Referee 2, as well as their helpful suggestions for improving the clarity and presentation of our work.

In the revised manuscript, we have addressed the comments by expanding the explanation of the intuition behind our method, providing a more detailed description of Step (c), and including a benchmark to illustrate the relevance of Step (b). We have also improved the figures for better readability. We believe these changes have strengthened the manuscript and made it more accessible to a broader audience.

We greatly appreciate the opportunity to revise our submission and thank the Editor for guiding the review process. We hope that the revised manuscript will now be found suitable for publication in SciPost Physics.

Sincerely,
Kohtaroh Sakaue, Hiroshi Shinaoka, Rihito Sakurai

---

## Round 2 · List of Changes

• Expanded the discussion on the intuition behind the method, particularly clarifying the roles and motivations of Steps (a) and (b).
  • Added a more detailed explanation of Step (c), including a description of the cost function, the optimization procedure, and its computational complexity.
  • Conducted and included a benchmark comparison to demonstrate the relevance and effectiveness of Step (b) (bond dimension compression).
  • Improved the clarity and presentation of Figure 5 by reducing data density and adjusting visual transparency.
  • Revised the introductory text before the formalization of the algorithm to provide better context and improve readability.
  • Performed minor edits throughout the manuscript to enhance clarity, accuracy, and consistency of notation.
  • Updated captions and figure labels for clarity and to avoid redundancy.

---

## Editorial Decision

published